# Impact of Hyponatremia on COPD Exacerbation Prognosis

**DOI:** 10.3390/jcm9020503

**Published:** 2020-02-12

**Authors:** María-Teresa García-Sanz, Sandra Martínez-Gestoso, Uxío Calvo-Álvarez, Liliana Doval-Oubiña, Sandra Camba-Matos, Carlos Rábade-Castedo, Carlota Rodríguez-García, Francisco-Javier González-Barcala

**Affiliations:** 1Emergency Department, Salnés County Hospital, 36600 Vilagarcía de Arousa, Spain; sandra.martinez.gestoso@sergas.es (S.M.-G.); liliana.doval.oubina@sergas.es (L.D.-O.); sandra.camba.matos@sergas.es (S.C.-M.); 2Respiratory Medicine Department, Hospital Arquitecto Marcide, 15405 Ferrol, Spain; uxio.calvo.alvarez@sergas.es; 3Respiratory Medicine Department, University Hospital Complex of Santiago de Compostela, 15706 Santiago de Compostela, Spain; carlos.rabade.castedo@sergas.es (C.R.-C.); carlota.rodriguez.garcia@sergas.es (C.R.-G.); francisco.javier.gonzalez.barcala@sergas.es (F.-J.G.-B.); 4Medicine Department, University Hospital Complex of Santiago de Compostela, 15706 Santiago de Compostela, Spain

**Keywords:** COPD, hyponatremia, prognosis

## Abstract

The most common electrolyte disorder among hospitalized patients, hyponatremia is a predictor of poor prognosis in various diseases. The aim of this study was to establish the prevalence of hyponatremia in patients admitted for acute exacerbation of chronic obstructive pulmonary disease (AECOPD), as well as its association with poor clinical progress. Prospective observational study carried out from 1 October 2016 to 1 October 2018 in the following hospitals: Salnés in Vilagarcía de Arousa, Arquitecto Marcide in Ferrol, and the University Hospital Complex of Santiago de Compostela, Galicia, Spain, on patients admitted for AECOPD. Patient baseline treatment was identified, including hyponatremia-inducing drugs. Poor progress was defined as follows: prolonged stay, death during hospitalization, or readmission within one month after the index episode discharge. 602 patients were enrolled, 65 cases of hyponatremia (10.8%) were recorded, all of a mild nature (mean 131.6; SD 2.67). Of all the patients, 362 (60%) showed poor progress: 18 (3%) died at admission; 327 (54.3%) had a prolonged stay; and 91 (15.1%) were readmitted within one month after discharge. Patients with hyponatremia had a more frequent history of atrial fibrillation (AF) (*p* 0.005), pleural effusion (*p* 0.01), and prolonged stay (*p* 0.01). The factors independently associated with poor progress were hyponatremia, pneumonia, and not receiving home O_2_ treatment prior to admission. Hyponatremia is relatively frequent in patients admitted for AECOPD, and it has important prognostic implications, even when mild in nature.

## 1. Introduction

Hyponatremia is the most common electrolyte disorder among hospitalized patients, particularly among the elderly and women, and in the presence of comorbidities [1,2,3,4,5,6]. Its prevalence varies from 8% to 51%, and it often occurs as an acute disease complication, as a chronic disease decompensation, or as a consequence of interventions during patient treatment [1,5,6]. Hyponatremia is a predictor of poor prognosis in various diseases, including heart failure, coronary syndrome, liver cirrhosis, chronic kidney disease, and stroke [3,5,7,8,9]. It is featured on clinical assessment scales, such as APACHE (Acute Physiology and Chronic Health disease classification system) [10] and PSI (Pneumonia Severity Index) [11]. However, whether hyponatremia has a direct effect on mortality, or whether it is a severity marker in certain patients, is unknown [3]. Prognostic factors in patients with acute exacerbations of chronic obstructive pulmonary disease (AECOPD) include baseline disease severity, frequency and severity of exacerbations, age, various comorbidities, history of previous admissions for AECOPD, severity of exacerbations, and a series of physiological and laboratory parameters, such as lung function, respiratory rate at admission, hypoxemia, and hypercapnia [12,13]. Although previous studies show that hyponatremia, particularly when severe, is a predictor of poor prognosis in chronic obstructive pulmonary disease (COPD) [2], few studies determine the prevalence and impact of hyponatremia on the prognosis in the patients with AECOPD requiring hospitalization.

The aim of this study was to establish the prevalence of hyponatremia in patients admitted for AECOPD, as well as its association with poor clinical progress, including prolonged stay, death during hospitalization, or readmission within one month after discharge.

## 2. Materials and Methods

Prospective observational study carried out from 1 October 2016 to 1 October 2018 in the following hospitals: Salnés in Vilagarcía de Arousa, Arquitecto Marcide in Ferrol, and the University Hospital Complex of Santiago de Compostela, Galicia, Spain, on the patients admitted for AECOPD who had agreed to participate and signed informed consent. COPD diagnosis, baseline severity, and exacerbation were defined according to the Global Initiative for Chronic Obstructive Pulmonary Disease (GOLD) criteria [14]. Baseline dyspnea was classified as per the extended Medical Research Council Dyspnoea (eMRCD) scale [15]. Comorbidity was assessed using the Charlson Index [16], and history of stroke, atrial fibrillation (AF), chronic heart failure (CHF), arterial hypertension (AHT), ischemic heart disease (IHD), cancer, cognitive impairment, chronic kidney disease (CKD), and anemia were recorded. Patients who had not smoked for over a year at the time of admission were considered former smokers [17]. Patients were divided into four groups by body mass index (BMI) as per the WHO standards: underweight (BMI < 18.5), normal weight (18.5 ≤ BMI < 25), overweight (25 ≤ BMI < 30), and obesity (BMI ≥ 30). Patient baseline treatment was identified, including hyponatremia-inducing drugs, such as thiazide diuretics, angiotensin-converting enzyme (ACE) inhibitors, angiotensin II receptor blockers (ARBs), non-steroidal anti-inflammatory drugs (NSAIDs), selective serotonin reuptake inhibitors (SSRIs), tricyclic antidepressants, carbamazepine, and lamotrigine. The use of at least 5 mg prednisone for three or more consecutive months was considered chronic corticosteroid treatment [18].

Vital signs, arterial blood gas, and chest X-rays were obtained upon patient arrival at the Emergency Department (ED). Complete blood count and serum biochemistry data obtained in both the ED and the hospitalization ward were recorded. Patients were considered to have hyponatremia at admission if serum sodium levels were below 135 mEq/L [2] in at least one of the tests performed from arrival at the ED to hospital discharge. Serum sodium was applied a correction factor of 2.4 for every 100 mg/dL increase in glucose concentration in patients with blood glucose >200 mg/dL [19]. Hyponatremia severity was categorized into 3 groups: mild (130–134 mmol/L), moderate (120–129 mmol/L) and severe (<120 mmol/L) [20]. Patients with hypernatremia, defined as serum sodium levels above 145 [21], were excluded from the analysis to avoid confounding factors between high serum sodium and poor progress. Patients with prolonged stay were identified as those with a stay equal to or greater than the median stay of the study population (in our case, 7 days) [22]. Causes of death were classified into 4 groups according to origin: (1) respiratory; (2) cardiovascular (IHD, CHF, and pulmonary embolism); (3) cancer; (4) other causes (septic shock, bronchial aspiration, unknown origin, …).

Poor progress was defined as follows: prolonged stay, death during hospitalization, or readmission within one month after the index episode discharge.

All the patients in the study signed an informed consent, and the study was approved by the Clinical Research Ethics Committee of Galicia (code 2016/460).

### Statistical Analysis

The data obtained through statistical analysis are expressed as mean values ± standard deviation (SD) in continuous variables, and as frequencies and percentages in categorical variables. Continuous variables were compared using the Student’s *t*-test or the Wilcoxon test; for categorical variables, the chi-square test and the Fisher’s exact test were used. Variables independently related to hyponatremia were identified by logistic regression, including those with *p* ≤ 0.05 in the univariate analysis. Variables associated with *p* < 0.05 were considered statistically significant. All analyses were completed with SPSS 15.

## 3. Results

A total of 602 patients were enrolled in the study; patient mean age was 73.8 years (SD 10.6) and 86% were male. 65 cases of hyponatremia (10.8%) were recorded, 53 of a mild nature (mean 132.6; SD 1.12), and 12 of a moderate nature (mean 126; SD 2.28). Of all the patients, 362 (60%) featured poor progress: 18 (3%) died at admission; 327 (54.3%) had a prolonged stay; and 91 (15.1%) were readmitted within one month after discharge. The most frequent cause of death was respiratory (53%). Baseline characteristics of the patients are shown in Table 1. Patients with worse prognosis were more frequently assigned GOLD Severe and Very Severe grades (*p* < 0.0001), had worse baseline dyspnea (*p* < 0.0001), higher percentage of flu vaccination, and higher AF (*p* 0.002), CHF (*p* 0.001), and CKD (0.002) frequency. Regular treatments are shown in Table 2. Of the patients, 47.7% received potentially hyponatremia-inducing treatments. Patients with poor progress were more frequently treated with short-acting bronchodilators (SABA (short-acting β2 agonists) (*p* 0.003); SAMA (short-acting anticholinergics) (*p* 0.04), chronic oral corticosteroids (*p* 0.005), theophylline (*p* 0.03), azithromycin (*p* 0.02), acetylcysteine (*p* 0.02), phosphodiesterase inhibitors (*p* 0.01), and home O_2_ (*p* 0.008), although these results were not maintained upon the multivariate analysis.

The most frequent cause of AECOPD had a viral origin, although that was more common in the patients with poor progress (*p* < 0.0001); they also demonstrated a higher heart rate (0.02), higher leukocytosis (*p* 0.04), higher neutrophilia (*p* 0.02), more cases of hyponatremia (*p* 0.005), and troponin elevation (*p* 0.01); higher urea, potassium, and PCO_2_ levels (*p* 0.001, *p* 0.01 and 0.001, respectively), higher frequency of pneumonia episodes (*p* 0.001), admission to the ICU (*p* 0.01), mechanical ventilation (*p* 0.02), and non-invasive mechanical ventilation (*p* 0.003). Patients with favorable progress had lower PO_2_ (*p* 0.009) (Table 3). The history of the patients with hyponatremia featured a higher frequency of AF (*p* 0.005), pleural effusion (*p* 0.01), higher leukocytosis (*p* 0.02), prolonged stay (*p* 0.01), and poor progress (*p* 0.01) (Table 4), and the relationship between hyponatremia and AF, pleural effusion and prolonged stay was maintained after the multivariate analysis (Table 5). The factors independently associated with poor progress were hyponatremia, pneumonia, and not receiving home O_2_ treatment prior to admission (Table 6).

## 4. Discussion

In our study, the prevalence of hyponatremia in patients admitted for AECOPD is 10.8% higher than that of patients with community-acquired pneumonia as reported by Cuesta et al. [1], but lower than that of patients hospitalized for different causes, including AECOPD [2,3,4,23,24]. In addition to the low prevalence of hyponatremia, the absence of moderate and severe cases also seems noteworthy, despite the high percentage of patients with risk factors to develop it (heart failure, chronic kidney disease, cancer, and certain treatments).

In our study, factors related to poor progress were hyponatremia, pneumonia, and not being home oxygen users. We also observed some differential characteristics between patients with hyponatremia and those with normal sodium levels: the former demonstrate higher prevalence of AF, chest X-ray with pleural effusion, and prolonged stay, and these associations were independent. The higher prevalence of AF in patients with hyponatremia could be explained by its arrhythmogenic capacity, appearing as a result of a series of changes in the electrical activity and properties of the sinus node and pulmonary veins [25,26]. Recent studies show pleural effusion on chest X-ray in 33%–50% of the patients admitted for acute heart failure, and pleural effusion associated with a high rate of adverse events after hospital discharge (readmission, new heart failure episode, or death) [27,28]. In our case, there was a history of heart failure in 30% of the patients, and pleural effusion was recorded in 12%. Although no relationship between pleural effusion and poor progress was found, almost 22% of the patients with hyponatremia also had pleural effusion. Compensatory mechanisms counteracting the low cardiac output include stimulation of the renin-angiotensin-aldosterone system, sympathetic nervous system and release of vasopressin, which contribute to an increase in blood volume and, therefore, to hyponatremia and pleural effusion [29].

Results of the previous studies with the same sodium level cutoff to define hyponatremia (< 135 mEq/L) are similar to ours. Chalela et al. associate hyponatremia with longer hospital stay and higher hospital mortality in patients admitted for AECOPD [2]. Al Mawed et al. report that hyponatremia is independently associated with hospital mortality and increase in dependence or need for care after discharge, regardless of the diagnosis leading to admission, including AECOPD [24]. The risk of readmission within the first 30 days after discharge due to an episode of heart failure was significantly higher in patients with persistent hyponatremia, regardless of the heart failure severity [30].

Pneumonia is an independent factor of poor prognosis in patients with COPD and has been associated with longer average stay and lower survival [31,32]. Apart from that, hyponatremia occurs in 8%–31% of adults with pneumonia, it can be mild and asymptomatic [33], and it is associated with adverse outcomes, such as higher hospital mortality, the need for intensive care, and longer average stay [1,34,35]. Causality of hyponatremia and pneumonia is uncertain, although the syndrome of inappropriate antidiuretic hormone (ADH) secretion seems to play an important role in this association [1]. Other contributing factors could include the loss of volume and sodium through lungs and skin as a result of hyperventilation and temperature in patients with pneumonia, or adrenal suppression due to corticosteroid treatment—even inhaled—frequent in patients with COPD [36].

In our study, continuous home oxygen therapy (HOT) reduces the risk of poor prognosis. Although HOT is known to increase survival, reduce admission rates and improve the quality of life in patients with COPD and chronic respiratory failure [37,38], the relationship between HOT and patient progress following an AECOPD episode is not clearly defined. Recent literature shows apparently contradictory results in the relationship between HOT and prognosis after AECOPD: mortality, admission rate and average stay increase according to some studies [39,40,41]; and patients with HOT would show higher survival rates after admission to ICU and lower admission numbers according to other authors [42,43].

Better progress of patients requiring HOT could be explained by stricter medical follow-up, allowing for base treatment optimization, health education strengthening, and therapy compliance improvement, consequently impacting progress in a beneficial manner [44,45,46]. It could also allow for early identification of exacerbations, optimizing early management [47,48,49] Patients with HOT have been observed to have lower acidosis during exacerbations –which is an indicator of the lower severity of acute episodes– than those not requiring HOT [42]. Other pathophysiological mechanisms –not sufficiently clarified yet– have also been identified, such as the impact on certain mitochondrial genes by HOT in patients with COPD, which could explain these differences in outcome [50].

This study has several limitations. The cause of hyponatremia was not established. The fact that we only had mild and moderate cases of hyponatremia did not allow us to verify the impact of sodium levels on clinical results. Treatments aimed at correcting hyponatremia were not evaluated. The observational nature prevents the study from establishing causality and excluding residual confounding factors.

In conclusion, it seems that hyponatremia is relatively frequent in patients admitted for AECOPD, and it has prognostic implications. Therefore, patients with hyponatremia should be carefully monitored.

## Figures and Tables

**Table 1 jcm-09-00503-t001:** Baseline characteristics of patients.

Characteristics	Good Progress	Poor Progress	*p*
	240 (40%)	362 (60%)	
Age	73.7 (SD 10.2)	73.9 (SD 10.9)	NS
Male	201 (83.8%)	317 (87.6%)	NS
Institutionalized	7 (3%)	11 (3%)	NS
FEV1%	52.9 (SD 18.1)	52.8 (SD 19.5)	NS
GOLD			<0.0001
1	33 (15.1%)	21 (6.5%)	
2	100 (45.7%)	114 (35.3%)	
3	51 (23.3%)	81 (24.9%)	
4	35 (16%)	109 (33.5%)	
Phenotype			NS
Emphysema exacerbator	37 (15.7%)	59 (17.2%)	
CB exacerbator	63 (26.8%)	119 (34.7%)	
Non-exacerbator	102 (43.4%)	125 (36.4%)	
Mixed	33 (14%)	40 (11.7%)	
BMI			NS
Underweight	37 (15.4%)	67 (18.2%)	
Normal weight	48 (20%)	81 (22.4%)	
Overweight	83 (34.6%)	123 (34.1%)	
Obesity	72 (30%)	90 (24.9%)	
Tobacco use			NS
Active smoker	74 (31.2%)	88 (25.7%)	
Former smoker	148 (62.4%)	226 (65.9%)	
Never smoker	15 (6.3%)	29 (8.5%)	
Baseline dyspnea			<0.0001
1	7 (3%)	12 (3.4%)	
2	68 (29%)	44 (12.6%)	
3	94 (40%)	94 (26.9%)	
4	46 (19.6%)	113 (32.4%)	
5a	18 (7.7%)	68 (19.5%)	
5b	2 (0.9%)	18 (5.2%)	
Flu-vaccinated	142 (59.2%)	255 (70.8%)	0.003
Charlson	6.1 (SD 26.1)	5.4 (SD 2.6)	NS
Stroke	21 (8.8%)	38 (10.5%)	NS
AF	48 (20%)	113 (31.2%)	0.002
CHF	53 (22.1%)	128 (35.4%)	0.001
AHT	134 (55.8%)	223 (61.6%)	NS
IHD	39 (16.3%)	70 (19.3%)	NS
DM	58 (24.2%)	88 (24.3%)	NS
Cancer	45 (18.8%)	93 (25.6%)	NS
Cognitive impairment	9 (3.8%)	26 (7.2%)	NS
CKD	19 (7.9%)	61 (16.9%)	0.002
Anemia	25 (10.4%)	57 (15.7%)	NS
Admissions previous year	0.7 (SD 1.2)	0.9 (SD 1.4)	NS
ED previous year	1.1 (SD 1.7)	1.2 (SD 2.0)	NS

AF: atrial fibrillation; AHT: arterial hypertension; BMI: body mass index; CB: chronic bronchitis; CHF: chronic heart failure; CKD: chronic kidney disease; DM: diabetes mellitus; ED: emergency department; FEV1%: Tiffeneau–Pinelli index; IHD: ischemic heart disease; NS: not significant

**Table 2 jcm-09-00503-t002:** Baseline treatment of patients.

Drug	Good Progress	Poor Progress	*p*
	240 (40%)	362 (60%)	
SABA	80 (33.3%)	164 (45.3%)	0.003
LABA	187 (77.9%)	281 (77.6%)	NS
IC	143 (59.6%)	239 (66%)	NS
Oral C.	6 (2.5%)	29 (8%)	0.005
SAMA	40 (16.7%)	85 (23.5%)	0.04
LAMA	158 (65.8)	240 (66.3%)	NS
Theophylline	17 (7.1%)	45 (12.4%)	0.03
Beta blocker	47 (19.6%)	60 (16.6%)	NS
Diuretic	94 (39.2%)	154 (42.5%)	NS
Azithromycin	3 (1.3%)	16 (4.4%)	0.02
Acetylcysteine	12 (5%)	37 (10.2%)	0.02
Antileukotrienes	5 (2.1%)	9 (2.5%)	NS
Phosphodiesterase inhibitors	5 (2.1%)	22 (6.1%)	0.01
Home O_2_	43 (17.9%)	99 (27.3%)	0.008
Hyponatremia-inducing drugs	119 (49.6%)	168 (46.4%)	NS

IC: inhaled corticosteroid; LABA: long-acting β2 agonists; LAMA: long-acting anticholinergics; NS: not significan; Oral C.: chronic oral corticosteroid; SABA: short-acting β2 agonists; SAMA: short-acting anticholinergics;

**Table 3 jcm-09-00503-t003:** Exacerbation characteristics.

Clinical Data	Good Progress	Poor Progress	*p*
	240 (40%)	362 (60%)	
Cause of Exacerbation			<0.0001
Viral	44 (18.3%)	26 (7.2%)	
Bacterial	168 (70%)	292 (80.7%)	
Non-infectious	28 (11.7%)	44 (12.2%)	
HR	93 (SD 19.7)	98 (SD 22.5)	0.02
SBP	138 (SD 25.8)	139 (SD 27.1)	NS
DBP	74 (SD 14.1)	76 (SD 16.0)	NS
T	36.7 (SD 0.9)	36.6 (SD 0.9)	NS
ICU	1 (SD 0.4)	13 (SD 3.6)	0.01
MV	2 (0.8%)	14 (3.9%)	0.02
NIMV	23 (9.6%)	67 (18.5%)	0.003
Leukocytes	10,831 (SD 4780)	11,675 (SD 5174)	0.04
Neutrophils (%)	75.2 (SD 13.8)	77.7 (SD 12.2)	0.02
Eosinophils (%)	2.2 (SD 13.5)	1.3 (SD 3.1)	NS
Glucose	135 (SD 53.3)	143 (SD 60.3)	NS
Urea	47 (SD 21.9)	55 (SD 33.9)	0.001
Creatinine	2.2 (SD 13.5)	1.3 (SD 3.1)	NS
Hyponatremia	15 (6.4%)	49 (13.6%)	0.005
K+	4.4 (SD 0.4)	4.5 (SD 0.8)	0.01
Albumin	3.8 (SD 0.4)	3.7 (SD 1.8)	NS
Fibrinogen	553 (105.8)	537 (123.4)	NS
CRP	54 (79.8)	68 (113.1)	NS
Troponin (+)	18 (8.7%)	51 (14.6%)	0.01
PO_2_	57 (SD 14.8)	62 (SD 29.0)	0.009
PCO_2_	44 (SD 10.6)	47 (SD 14)	0.001
CO_3_H^−^	28.2 (SD 6.1)	28.6 (SD 5.5)	NS
PH	7.4 (SD 0.0)	7.3 (SD 16.2)	NS
Pneumonia XR	37 (15.5%)	96 (26.8%)	0.001
Pleural effusion XR	23 (9.7%)	50 (14%)	NS

CRP: C-reactive protein; DBP: diastolic blood pressure; HR: heart rate; ICU: intensive care unit; K+: potassium; MV: mechanical ventilation; NIMV: non-invasive mechanical ventilation; NS: not significant; SBP: systolic blood pressure; T: temperature; XR: chest X-ray.

**Table 4 jcm-09-00503-t004:** General and clinical characteristics of patients with and without hyponatremia.

Variables	WITHOUT Hyponatremia	WITH Hyponatremia	*p*
	535 (89%)	65 (11%)	
Age n (SD)	73.9 (10.6)	73.4 (10.6)	NS
Male	460 (86%)	56 (86.2%)	NS
BMI n (SD)	28.5 (5.8)	27.4 (4.7)	NS
FEV1% n (SD)	52.5 (18.9)	55.6 (19.7)	NS
Comorbidities			
AF	134 (25%)	27 (41.5%)	0.005
Cognitive impairment	33 (6.2%)	0 (0.0%)	0.04
Anemia	68 (12.7%)	14 (21.5%)	0.05
Hyponatremia-inducing drugs	249 (46.5)	37 (56.9%)	NS
Admissions previous year n (SD)	0.8 (1.3)	0.9 (1.6)	NS
Exacerbations previous year	1.2 (1.8)	1.4 (2.5)	NS
Pleural effusion XR	59 (11.1%)	14 (21.9%)	0.01
Leukocytes n (SD)	11,160 (4897)	12,623 (5878)	0.02
CRP n (SD)	56 (82.7)	109 (185.4)	0.05
Hospital mortality	14 (2.6%)	3 (4.6%)	NS
Prolonged stay	282 (52.7%)	45 (69.2%)	0.01
Readmission within 30 days	84 (15.7%)	7 (10.8%)	NS
Poor progress	313 (58.5%)	48 (73.8%)	0.01

AF: atrial fibrillation; BMI: body mass index; CRP: C-reactive protein; FEV1%: Tiffeneau–Pinelli index; NS: not significant; XR: chest X-ray.

**Table 5 jcm-09-00503-t005:** Hyponatremia-related factors.

Multivariate Analysis			
Variables	Hazard Ratio	95% CI	*p*
AF	1.95	(1.12; 3.41)	0.01
Pleural effusion	2.01	(1.03; 3.94)	0.04
Prolonged stay	1.79	(1.01; 3.15)	<0.0001

AF: atrial fibrillation. Adjusted by: AF, Anemia, Cognitive impairment, Pleural effusion XR, Leukocytes, CPR, Prolonged stay, Poor progress.

**Table 6 jcm-09-00503-t006:** Factors related to poor progress following admission for AECOPD.

Multivariate Analysis			
Variables	Hazard Ratio	95% CI	*p*
Home O_2_	0.35	(0.16; 0.74)	0.006
Pneumonia	1.84	(1.05; 3.24)	0.03
Hyponatremia	3.68	(1.57; 8.58)	0.003

Adjusted by: Etiology of exacerbation, Mechanical ventilation, Non-invasive mechanical ventilation, Flu-vaccinated, GOLD, Baseline dyspnea, AF, CHF, CKD, Baseline treatment (SABA, Oral corticosteroids, SAMA, Theophylline, Azithromycin, Acetylcysteine, Phosphodiesterase inhibitors), Home O_2_, Pneumonia XR, Positive Troponin, Hyponatremia, HR, Admission at ICU, Leukocytes, Neutrophils (%), Urea, Potassium, PO_2_, PCO_2_.

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
