# Peer review of "Impact of Hyponatremia on COPD Exacerbation Prognosis"

_jcm, 2020, doi:10.3390/jcm9020503_

Round 1

Reviewer 1 Report

This is a multicentre prospective cohort study that evaluated the prevalence and significance of hyponatramia in 602 patients with acute exacerbation of COPD. The author concluded that hyponatraemia is a relatively common feature in patients with acute exacerbation of COPD and associated with prolonged hospitalisation.

Major comments

1. Abstract, Line 27: “all of mild nature”. This line is unclear needs to be expanded, with the provision of mean +/- standard deviation of sodium levels.

2. The Result section needs to be expanded, only little information provided currently.

• The degree of hyponatraemia needs to be provided in details, including mean +/- standard deviation and the number of patients in each group of hyponatraemia severity

• Provide a summary of the main findings for each table included.

3. It is unclear whether there is a relationship between the severity of hyponatraemia and poor clinical progress.

4. Table 4: It is unclear which variables were included in the univariate analyses? Only those in Table 4, or all baseline characteristics and treatment. If it is the later, all results should be provided in Table 4. 5. Discussion, Line 119: It should be “not being home oxygen users” instead of “being home oxygen users”.

Minor comments

1. Abstract, Line 19-20: Remove “including prolonged stay, death during hospitalization or readmission within the first month after discharge” – this is repetitive and has been included in the Methods part.

2. Introduction, Line 38-41: The first sentence of the introduction needs to be restructured into two sentences to improve readability.

3. Provide the full terms of all acronym at first appearance in the manuscript

4. Method, Line 85: Remove “…” 5. Conclusion, Line 164: Remove “we can say that”

Author Response

REVIEWER 1

Major comments

Abstract, Line 27: “all of mild nature”. This line is unclear needs to be expanded, with the provision of mean +/- standard deviation of sodium levels.

Abstract modified to provide +/- standard deviation of sodium levels.

The Result section needs to be expanded, only little information provided currently.

The degree of hyponatraemia needs to be provided in details, including mean +/- standard deviation and the number of patients in each group of hyponatraemia severity

          Done.

Provide a summary of the main findings for each table included.

          Done.

It is unclear whether there is a relationship between the severity of hyponatraemia and poor clinical progress.

In this study there are just 65 cases of hyponatremia: 53 of a mild nature, 12 of a moderate nature, none severe. We did not make this test because there are cohorts with few cases to deduce strong conclusions.

Table 4: It is unclear which variables were included in the univariate analyses? Only those in Table 4, or all baseline characteristics and treatment. If it is the later, all results should be provided in Table 4.

Findings in Tables 1,2 and 3 are the result of univariate analysis. Multivariate analysis in table 6 includes all parameters from Tables 1, 2 and 3 with p < 0.05, as explained in the Method section. Only “pneumonia” and “hyponatremia” from Table 3, and “Home O2” from Table 2 maintained its significance after multivariate analysis (adjusted by Exacerbation_Etiology, MV, NIMV, Vaccinated, GOLD, Baseline_Dyspnea, AF, CHF, CKD, Baseline_SABA, Baseline_Oral C., Baseline_SAMA, Baseline_Theophylline, Baseline_Azithromycin, Baseline_Acetylcysteine, Home_O2, Pneumonia_XR, Positive_Troponin, Phosphodiesterase_Inhib, Hyponatremia, HR, ICU, Leukos, Neutros_percent, Urea, Potassium, PO2, PCO2).

Table 4 is intended to show some differences between patients with and without hyponatremia, in spite of that is not the aim of this study. The table 4 shows all variables included. The table 5 includes all parameters of table 4 with p < 0.05.

Discussion, Line 119: It should be “not being home oxygen users” instead of “being home oxygen users”.

Correct. The paragraph has been amended.

Minor comments

Abstract, Line 19-20: Remove “including prolonged stay, death during hospitalization or readmission within the first month after discharge” – this is repetitive and has been included in the Methods part.

Done.

Introduction, Line 38-41: The first sentence of the introduction needs to be restructured into two sentences to improve readability.

          Done.

Provide the full terms of all acronym at first appearance in the manuscript

Done.

Method, Line 85: Remove “…” 5. Conclusion, Line 164: Remove “we can say that”

“We can say” has been rephrased as “it seems” in the conclusion.

Reviewer 2 Report

This manuscript of Teresa Garcia-Sanz Maria et al presents the results of a prospective observational study on the relationship of AECOPD progress and hyponatremia.

The results are not very novel and there are significant methodological , interpretational issues.

Major criticism.

It is not clear if the parameters of poor progress were taken as individual factors or in sum.(Tables1,3,4). It is obvious that prolong hospital stay and death have significant effects on poor prognosis. Thus, the classification of the subjects as good /poor progress lucks solid scientific merit.

2 .Table 3 reveals a number of parameters reaching statistical significance between good and poor progress patients such us : leukocyte counts ,K+, troponin, pneumonia among others. These findings may have affected the results and the conclusions.

Table 4, apart the presence of hyponatremia or not , shows differences in leukocytes, CRP, anemia. This rise the question that patients with hyponatremia may have also some kind of infection affecting more their poor progress than this mild hyponatremia! In Table 4, prolonged hospital stay and Poor progress were presented as different factors. Since this is a prospective study why the authors did not measure Atrial natrioduretic peptide or/and angiotensin II in order to verify the pathogenesis of hyponatremia?? Was any effort to treat hyponatremia during the hospital stay? It is not clear when hyponatremia was detected on admission or during hospital management? It would be better to study only those with hyponatremia on admission.

Minor comments.

Title: Hypotermia is not hyponatremia!! Although, inappropriate secretion of ADH is mentioned in the discussion, no clinical sighs of this syndrome were shown. The discussion on HOT is confusing.

4 The conclusion is very weak as far the “important prognostic implication” is concerned.

Author Response

 REVIEWER 2

The results are not very novel and there are significant methodological , interpretational issues.

The results of this study provide information on a relatively frequent aspect --but not well known in COPD exacerbation, and often undervalued (Peri A, Grohé C, Berardi R, Runkle I. SIADH: differential diagnosis and clinical management. Endocrine. 2017 Jan;55(1):311-319)

We considered it relevant, since patients with mild hyponatremia were included, and the impact on the prognosis has been statistically significant. We consider it to be clinically relevant, since according to the data in Table 6, hyponatremia triples the probability of a poor clinical progress. Also, the worse prognosis seems easily detectable, since serum Na is a usual, low-cost determination parameter.

The information available is scarce and has some limitations:Chalela (ref 2): Single-center study on patients above 40; Al mawed (ref 24): Healthcare data base data. Medical records were not reviewed to confirm diagnostic accuracy. It includes various diseases --not just COPD: De Vecchis (ref 30): On heart failure.

Major criticism.

It is not clear if the parameters of poor progress were taken as individual factors or in sum.(Tables1,3,4). It is obvious that prolong hospital stay and death have significant effects on poor prognosis. Thus, the classification of the subjects as good /poor progress lucks solid scientific merit.

          The parameters were taken in sum, as explained in the Method section.

Table 3 reveals a number of parameters reaching statistical significance between good and poor progress patients such us: leukocyte counts ,K+, troponin, pneumonia among others. These findings may have affected the results and the conclusions.

These findings in Table 3 are the result of univariate analysis. Multivariate analysis includes all parameters from Tables 1, 2 and 3 with p < 0.05, as explained in the Method section. Only “pneumonia” and “hyponatremia” from Table 3, and “Home O2” from Table 2 maintained its significance after multivariate analysis (adjusted by Etiology of exacerbation, Mechanical ventilation, Non-invasive mechanical ventilation, Flu-vaccinated, GOLD, Baseline dyspnea, AF, CHF, CKD, Baseline Treatment (SABA, Oral corticosteroid., SAMA, Theophylline, Azithromycin, Acetylcysteine, phosphodiesterase inhibitor), Home_O2, Pneumonia_XR, Positive_Troponin, Hyponatremia, HR, Admission at ICU, Leukocytes, Neutrophils(%), urea, Potassium, PO2, PCO2) .

When doing a multivariate analysis, the impact modifying the effect of all other covariates is already taken into account.

Table 4, apart the presence of hyponatremia or not, shows differences in leukocytes, CRP, anemia. This rise the question that patients with hyponatremia may have also some kind of infection affecting more their poor progress than this mild hyponatremia!

As seen in Table 3, patients with a viral infection have a better prognosis, and those with a bacterial infection have a worse prognosis, considering the univariate analysis only.Once included in the multivariate analysis, the type of infection does not maintain a significant impact on patient progress.

In Table 4, prolonged hospital stay and Poor progress were presented as different factors.

Table 4 is intended to show some differences between patients with and without hyponatremia, including poor progress (as an indicator combining various factors: …x y z ….:  as explained in the Method section) and its parameters taken as individual factors.

Since this is a prospective study why the authors did not measure Atrial natrioduretic peptide or/and angiotensin II in order to verify the pathogenesis of hyponatremia??

Was any effort to treat hyponatremia during the hospital stay?

These parameters are not usually determined in the usual clinical practice in the hospitals participating in the study.

In the usual clinical practice in our hospitals, we try to correct hyponatremia when detected, particularly in the most serious cases, but these data have not been collected in our study. The aim of the study was assessing the impact of the presence of hyponatremia --at any time during hospital admission-- on the progress of patients in routine clinical practice, regardless of the measures taken to correct hyponatremia. The following has been added to the limitations of the study: treatments aimed at correcting hyponatremia have not been evaluated.

It is not clear when hyponatremia was detected on admission or during hospital management? It would be better to study only those with hyponatremia on admission.

As explained in the Method section, “patients were considered to have hyponatremia at admission if showing serum sodium levels below 135 mEq/L in at least one of the tests done from arrival at the ED to hospital discharge”.

Minor comments.

Title: Hypotermia is not hyponatremia!!

Amended.

Although, inappropriate secretion of ADH is mentioned in the discussion, no clinical sighs of this syndrome were shown.

Inappropriate secretion of ADH is mentioned as possible causality of hyponatremia and pneumonia, but it is not the topic of this study.

The clinical signs of hyponatremia are quite nonspecific and relatively frequent, particularly in elderly patients requiring hospitalization, regardless of the etiology ((Peri A, Grohé C, Berardi R, Runkle I. SIADH: differential diagnosis and clinical management. Endocrine. 2017 Jan;55(1):311-319; Grant P, Ayuk J, Bouloux PM, Cohen M, et al. The diagnosis and management of inpatient hyponatraemia and SIADH. Eur J Clin Invest. 2015 Aug;45(8):888-94). Thus, and since these symptoms were not specifically searched considering the real-life nature of this study, attributing patient symptoms to hyponatremia is difficult. This is one of the aspects that makes highlighting the importance of hyponatremia and its observation in analytical determinations relevant, given the wide availability and low cost of the latter.

The discussion on HOT is confusing.

The discussion of HOT shows the apparently contradictory results in the relationship between HOT and prognosis after AECOPD according to different authors. We modify the HOT paragraph to improve the understanding of the text.

The conclusion is very weak as far the “important prognostic implication” is concerned.

“We can say” has been rephrased as “it seems” in the conclusion. We believe that this is more correct, since --although hyponatremia triples the probability of poor progress (which seems relevant)-- we understand that a single study, even if multicenter and with a relatively large sample size, does not seem enough to state that the findings are conclusive.

Round 2

Reviewer 1 Report

Thank you for the revised manuscript. Although the authors claimed to have made changes based on the comments in the response, the following ones have not yet been addressed in the actual revised manuscript.

Abstract, Line 19-20: Remove “including prolonged stay, death during hospitalization or readmission within the first month after discharge” – this is repetitive and has been included in the Methods part. Result section: The degree of hyponatraemia needs to be provided in details, including mean +/- standard deviation and the number of patients in each group of hyponatraemia severity Discussion, Line 119: It should be “not being home oxygen users” instead of “being home oxygen users”.

Additional typo noted in the revised manuscript:

Line 40: “Hiponatremia” Line 194: “shoul be”

Author Response

Abstract, Line 19-20: Remove “including prolonged stay, death during hospitalization or readmission within the first month after discharge” – this is repetitive and has been included in the Methods part.

Done

Result section: The degree of hyponatraemia needs to be provided in details, including mean +/- standard deviation and the number of patients in each group of hyponatraemia severity

Done

Discussion, Line 119: It should be “not being home oxygen users” instead of “being home oxygen users”.

Done

Additional typo noted in the revised manuscript:

Line 40: “Hiponatremia”

Amended.

Line 194: “shoul be”

Amended.